# Diagnostic Yield of Exome Sequencing in Fetuses with Sonographic Features of Skeletal Dysplasias but Normal Karyotype or Chromosomal Microarray Analysis: A Systematic Review

**DOI:** 10.3390/genes14061203

**Published:** 2023-05-30

**Authors:** Kai Yeung Tse, Ilham Utama Surya, Rima Irwinda, Kwok Yin Leung, Yuen Ha Ting, Ye Cao, Kwong Wai Choy

**Affiliations:** 1Department of Obstetrics and Gynaecology, Queen Elizabeth Hospital, Hong Kong SAR, China; barontse@hotmail.com; 2Department of Obstetrics and Gynaecology, Faculty of Medicine, Universitas Indonesia, Dr. Cipto Mangunkusumo Hospital, Jakarta 10430, Indonesia; ilhamutama@yahoo.com (I.U.S.); rima.irwinda@yahoo.com (R.I.); 3Gleneagles Hospital Hong Kong, Hong Kong SAR, China; ky@kyleung.org; 4Department of Obstetrics and Gynaecology, Faculty of Medicine, The Chinese University of Hong Kong, Hong Kong SAR, China; tingyh@cuhk.edu.hk; 5Department of Obstetrics and Gynaecology, Prince of Wales Hospital, Shatin, N.T., Hong Kong SAR, China; 6Shenzhen Research Institute, The Chinese University of Hong Kong, Shenzhen 518057, China

**Keywords:** whole-exome sequencing, CMA, prenatal diagnosis, skeletal dysplasia, systematic review

## Abstract

Skeletal dysplasias are a group of diseases characterized by bone and joint abnormalities, which can be detected during prenatal ultrasound. Next-generation sequencing has rapidly revolutionized molecular diagnostic approaches in fetuses with structural anomalies. This review studies the additional diagnostic yield of prenatal exome sequencing in fetuses with prenatal sonographic features of skeletal dysplasias. This was a systematic review by searching PubMed for studies published between 2013 and July 2022 that identified the diagnostic yield of exome sequencing after normal karyotype or chromosomal microarray analysis (CMA) for cases with suspected fetal skeletal dysplasias based on prenatal ultrasound. We identified 10 out of 85 studies representing 226 fetuses. The pooled additional diagnostic yield was 69.0%. The majority of the molecular diagnoses involved de novo variants (72%), while 8.7% of cases were due to inherited variants. The incremental diagnostic yield of exome sequencing over CMA was 67.4% for isolated short long bones and 77.2% for non-isolated cases. Among phenotypic subgroup analyses, features with the highest additional diagnostic yield were an abnormal skull (83.3%) and a small chest (82.5%). Prenatal exome sequencing should be considered for cases with suspected fetal skeletal dysplasias with or without a negative karyotype or CMA results. Certain sonographic features, including an abnormal skull and small chest, may indicate a potentially higher diagnostic yield.

## 1. Introduction

Skeletal dysplasias are a group of rare genetic disorders involving abnormal development, growth, and maintenance of the human skeleton system. Skeletal dysplasia is defined by the Fetal Medicine Foundation as either (1) shortening of long bones (usually regarded as at least less than 2 standard deviations), (2) abnormal shape of long bones, (3) reduced echogenicity of bones, and/or (4) absence of extremities [1]. The prevalence of skeletal dysplasia is about 2.4/10,000 births and 9.1/1000 among perinatal deaths; the most common forms of skeletal dysplasias detected in prenatal ultrasound are thanatophoric dysplasia, achondroplasia, osteogenesis imperfecta, and achondrogenesis [2,3]. According to nosology and the classification of genetic skeletal disorders, there are 461 different diseases, which are classified into 42 groups and involving 437 different genes [4].

Several large cohort studies evaluating the incremental diagnostic yield of CMA in fetuses with prenatal ultrasound anomalies suggested that it was around 6% [5,6,7]. Two meta-analyses identified an increased diagnostic yield of 7–10% over karyotype in pregnancies with structural fetal abnormalities [8]. The American College of Obstetricians and Gynecologists (ACOG) now recommends CMA as the first-tier test for fetal structural anomalies.

However, skeletal dysplasias do comprise many monogenic disorders due to nucleotide level changes. Therefore, CMA cannot detect such pathogenic variants. With the advancement of next-generation sequencing and the reduction in costs, numerous studies started to investigate the utility of whole-exome sequencing (ES) or even whole-genome sequencing in diagnosing these monogenic disorders. Exome sequencing is a DNA-based sequencing method that targets almost all the exons in the genome [9]. The exome makes up about 1.5% of the genome, primarily exons of genes that code for protein. Two of the landmark prenatal exome sequencing papers by Lord et al. and Petrovski et al. have tried to include a large number of prenatal samples (610 and 517, respectively) with fetal structural anomalies and to determine the additional diagnostic yield for various anomalies [10,11]. They demonstrated that WES had an additional diagnostic yield of 15.4–24% for those with skeletal anomalies.

Recently, the International Society for Prenatal Diagnosis (ISPD) updated its position statement for the indication of applying genome sequencing (including exome sequencing) in the prenatal setting. First, they suggested that prenatal sequencing may be offered to those with a fetus having a single major anomaly or multiple organ system anomalies, either with a normal CMA but the genetic expert review considers the phenotype suggestive of a possible genetic etiology, or when there are multiple organ system anomalies strongly suggesting a single gene disorder, even without prior genetic testing [12].

However, given the rarity of the disorder, most studies only include a small sample size in evaluating the additional diagnostic yield for exome sequencing in prenatal skeletal dysplasias. Moreover, given the wide range of sonographic features of skeletal dysplasias, it could be very difficult for clinicians to select cases where there is a higher probability for additional diagnostic yield. This systemic review aims to combine different studies to achieve a larger cohort in determining the additional diagnostic yield and to determine sonographic features of skeletal dysplasias that pertain to a higher diagnostic yield.

## 2. Materials and Methods

### 2.1. Eligibility Criteria

Following PRISMA guidelines, a systematic literature review was performed. Studies were included in this review if they met the following criteria: (i) retrospective or prospective cohorts of pregnancies undergoing ES (whole, clinical, or targeted) or WGS for diagnosis of skeletal dysplasias; (ii) CMA/karyotype was negative or nondiagnostic; (iii) testing was initiated based on the prenatal sonographic phenotype; and (iv) full-text report was available in English language. Skeletal dysplasias are defined as a fetus reported involving abnormal development, growth, and maintenance of the human skeleton system. The criteria are defined by the Fetal Medicine Foundation as either (1) shortening of long bones (usually regarded as at least less than 2 standard deviations), (2) abnormal shape of long bones, (3) reduced echogenicity of bones, and/or (4) absence of extremities.

The aim of this review was to focus on the prenatal diagnosis of skeletal dysplasia. Therefore, we would only extract prenatal cases suspected of fetal skeletal dysplasia in those WES studies mixed with prenatal and postnatal skeletal dysplasia for this analysis. Similarly, for studies with mixed indications for testing (e.g., some cases initiated because of fetal phenotype and some because of family history), we only included studies where fetal anomaly cases could be extracted. Studies in which ES was completed postnatally for some or all cases, but had been initiated based on the prenatal phenotype alone, were included. Variants classified as pathogenic/likely pathogenic according to diagnostic-standard guidelines and determined to be causative of the fetal phenotype were considered in diagnostic yield. We also excluded case reports, series, reviews, editorials, and commentaries.

### 2.2. Information Sources and Strategy Searches

Electronic searches on PubMed were conducted for records published between 2013 and July 2022. We searched the PubMed database for ((prenatal) AND (exome)) AND (skeletal) or ((prenatal) AND (exome)) AND (structural). Keywords with word variants of the terms ‘exome sequencing’ and ‘prenatal’ were used in an attempt to screen for more relevant studies. This search strategy is available from the corresponding author on request. This data range was chosen because prenatal ES was not used prior to 2013. This search was initially conducted on 14 July 2022. Additional potentially relevant studies were identified by manually searching reference lists of relevant studies and published reviews and conference proceedings of prenatal and genetics conferences of relevant major societies in the last 3 years. Search terms were variations on the keywords ‘prenatal diagnosis’ and ‘exome sequencing’.

### 2.3. Study Selection

After the removal of duplicates, two reviewers (I.U.S. and Y.C.) independently screened titles and abstracts. For abstracts identified as potentially relevant, full-text articles were retrieved and reviewed against the inclusion and exclusion criteria. Any disagreement between reviewers was resolved by discussion. Where the same data were presented more than once, the most recent study was selected.

### 2.4. Data Extraction

The following data, where available, were extracted by two reviewers (I.U.S. and Y.C.) into a datasheet: study setting, sample size, study inclusion criteria, ES approach and its platform, prenatal sonographic phenotypes used for interpretation, number of fetuses with diagnostic variants, variants of uncertain significance, incidental findings, gestation at testing, test turnaround time, pregnancy outcomes, and impact on management. For studies performed with CMA in parallel with sequencing, the cases with negative CMA were extracted in order to be comparable with other studies where chromosomal abnormalities were ruled out prior to ES.

### 2.5. Data Synthesis

The primary outcome of interest was the incremental diagnostic yield of ES (i.e., the proportion of cases in which a diagnostic genetic variant is detected with ES). In individual studies, some of them included all VUS identified as diagnostic. We adopted stricter criteria in determining cases that were diagnosed with ES. Cases with VUS identified were not regarded as diagnostic. Cases where one pathogenic/likely pathogenic variant for an autosomal recessive disease was found, but the second mutation was not found or only a VUS was found, were not regarded as diagnostic.

### 2.6. Subgroup Analysis

Various sonographic features of skeletal dysplasias could be identified prenatally. The ultrasound features were extracted from individual studies and their supplementary files. Features that were identified postnatally or during postmortem examination were not included. The different features were then grouped into two major categories, which are (1) isolated short long bones—where short long bones were the only feature being described, with no suspicion of other features of skeletal dysplasias and (2) nonisolated short long bones (this category includes all cases with short long bones plus other sonographic features suggestive of skeletal dysplasias).

To look further into individual sonographic features of skeletal dysplasias and the respective clinical implications, subgroup analysis was also performed according to the described sonographic features. These features included:Abnormal curvature of long bones;Suspected fracture of bones, including those with angulated long bones;Reduced or abnormal ossification of bones;Absent long bones (including radius, tibia, etc.), absent or abnormal phalanges;Abnormal joint posture, including talipes, contractures;Abnormal skull, including abnormal skull shape, macrocephaly;Abnormal facial profile, including flattened face, absent nasal bone, retro/micrognathia;Small chest, including bell-shaped thorax, small chest circumferenceAbnormal spine, including scoliosis;Hydropic features, including cystic hygroma, subcutaneous edema, pleural effusion.

Another subcategory was multisystem anomalies, where sonographic features of other major system anomalies were found. These include neural tube defects, cardiac anomalies, and renal dysplasia. Soft markers (including echogenic foci of left ventricles and echogenic bowel) were not included as multisystem anomalies. Where only the presumptive prenatal diagnosis (such as “achondroplasia”) was mentioned in the case description, but no actual description of the sonographic features was made, these are not included in the subgroup analysis.

## 3. Results

### 3.1. Characteristics of This Study Cohort

A total of 85 studies were found in the initial search from PubMed. Overall, eight reviews and 47 case reports were screened out. Of the studies included, 15 focused only on a single condition rather than skeletal dysplasias as a whole, and 5 studies did not include prenatal samples and were also screened out. A total of 10 cohort studies were found. These included a total of 230 cases with a prenatal suspicion of skeletal dysplasias. Among these, two cases were found to have trisomy 18, and two cases were found to have microdeletions. After excluding these 4 cases, all the remaining 226 cases underwent exome sequencing after an initial negative karyotype or CMA (Figure 1).

Chandler et al. (16 cases) was the only study that was performed in the UK. All the other studies (a total of 210 cases) were performed in China. The mean gestational age at diagnosis of skeletal dysplasia with USG was 24.3 weeks (range 12–40 weeks), of which 123 out of the 226 cases were diagnosed before 24 weeks of gestation. A total of 108 cases underwent termination of pregnancy (TOP). There were twelve live births, one stillbirth, one neonatal death, and three cases that were ongoing pregnancies at the time of the study drafting. The remaining 101 cases did not have their pregnancy outcomes described, as this was not included as a part of the original research protocol.

### 3.2. Pooled Diagnostic Yield with ES

The overall additional diagnostic yield of ES in the setting of negative karyotype or CMA was 69.0% (156/226) (Table 1). Of the 156 cases with molecular diagnoses, 134 (85.9%) were due to monoallelic/heterozygous variants, while 22 (14.1%) were due to homozygous/compound heterozygous variants. There were 72% due to de novo variants, and 8.7% were due to variants inherited from parental origin. The rest of the cases either did not perform parental testing, or the results of parental analysis were not mentioned in the studies. There were a total of 37 genes encompassing all 156 diagnosed cases. The most commonly recurring four genes were *FGFR3*, *COL1A1*, *COL1A2,* and *COL2A1,* which accounted for 70.7% of the cases (111/157). Detailed phenotype–genotype correlation and the molecular diagnosis can be found in the Appendix A.

### 3.3. Subgorup Analysis—Isolated and Nonisolated Short Bone

A total of 43 cases had isolated short long bones. Of these, 29 cases had a pathogenic or likely pathogenic variant identified with ES, contributing to an additional diagnostic yield of 67.4% (Figure 2). The gene harboring most of the pathogenic variants in most of the cases was *FGFR3,* comprising 58.6% of the diagnosed cases. *COL1A2* was the next commonest, comprising 13.8% of the diagnosed cases. Of all the cases, 14 (32.6%) resulted in TOP, and 2 (4.7%) were live births.

A total of 136 cases had short long bones reported with additional features, of which 105 cases had a pathogenic or likely pathogenic variant identified with ES, giving an additional diagnostic yield of 77.2% (Figure 2). The four most recurring genes (*FGFR3*, *COL1A1*, *COL1A2*, and *COL2A1*) comprised 76.2% of diagnosed cases. Other recurring genes included *ALPL*, *IFITM5*, *LBR*, *EBP,* and *DYNC2H1*. There were 93 cases (68.4%) undergoing TOP. There were 10 live births (7.3%), 1 stillbirth (0.7%), 1 neonatal death (0.7%), and 1 ongoing pregnancy.

Altogether, there were 179 cases that demonstrated short long bones. Among these, 134 cases (74.9%) had positive findings with ES. There was no statistically significant difference (*p* = 0.198) found in the additional diagnostic yield between the isolated and nonisolated short long bone subgroups.

### 3.4. Subgroup Analysis: Various Phenotypes for Skeletal Dysplasia (Figure 3)

#### 3.4.1. Abnormal Curvature

There were 72 cases with abnormal curvature of bones. Among the 55 cases (76.3%) with variants diagnosed with ES, *FGFR3* and *COL1A1* were the two commonest genes, each comprising 30.9% of diagnosed cases. A total of 42 cases (58.3%) underwent TOP; 7 were live birth (9.7%), and 2 were ongoing pregnancies.

#### 3.4.2. Suspected Fracture or Angulated Long Bones

There were a total of 12 cases; 9 cases (75%) had positive findings with ES, with *COL1A1* being the most recurrent (33% of all diagnosed cases). A total of 6 cases (50%) underwent TOP, and there was one live birth and one ongoing pregnancy.

#### 3.4.3. Reduced or Abnormal Ossification

A total of seven cases had reduced or abnormal ossification of bones. Five cases (71%) had positive findings with ES, including the *COL1A1*, *COL2A1*, *EBP,* and *PPIB* genes. Four cases (57.1%) underwent TOP, and one had neonatal death.

#### 3.4.4. Absent Long Bone, Absent or Abnormal Phalanges

Among the 20 cases, only 6 cases (30%) had additional findings with ES, with five different gene variants identified. Six cases (30%) underwent TOP.

#### 3.4.5. Abnormal Joint Posture, Contractures

There were a total of 31 cases; 14 cases (45%) had additional findings with ES, scattering across different genes. *COL1A1* was the most recurring gene (21.4%). There were 12 cases of TOP (38.7%), 2 live births, and 1 ongoing pregnancy.

#### 3.4.6. Abnormal Skull

Among the 24 cases with an abnormal shape of the skull or macrocephaly, 20 cases had a positive yield with ES (83.3%). *FGFR3* comprised 12 cases (60%) of them. A total of 19 cases (79.2%) underwent TOP, and there was 1 case of live birth and 1 case of stillbirth.

#### 3.4.7. Abnormal Face

A total of 17 cases had an abnormal face, including retrognathia/micrognathia, absent nasal bone, and flattened facial profile, of which 9 cases (52.9%) had positive findings with ES. A total of 10 cases (58.8%) underwent TOP.

#### 3.4.8. Small Chest

A total of 57 cases had a small chest, of which 47 cases (82.5%) had additional findings with ES. *FGFR3* was the commonest gene variant found (20 cases, 42.6%), followed by *COL1A1* (11 cases, 23.4%). A total of 44 cases underwent TOP (77.2%). One was a live birth, one resulted in neonatal death, and two were ongoing pregnancies.

#### 3.4.9. Abnormal Spine

A total of 12 cases had an abnormal spine. Seven cases (58.3%) had positive findings with ES. Five cases (41.7%) had undergone TOP.

#### 3.4.10. Hydropic Features

A total of 13 cases had shown hydropic features, including cystic hygroma, subcutaneous edema, and pleural effusion. Eight cases (61.5%) had additional variants identified with ES. Seven cases (53.8%) went for TOP, and two were live birth.

#### 3.4.11. Multisystem Anomalies

A total of 27 of the cases had predominant features of skeletal dysplasias combined with other system abnormalities (including ventriculomegaly, ventricular septal defect, cystic renal dysplasia, and spina bifida), of which 18 cases (66.7%) had positive findings with ES. *FGFR3* was the commonest gene identified (five cases, 27.8%). Of the total cases, 14 underwent TOP (51.9%). The summary and comparison of cases with different sonographic features, and the additional diagnostic yield with ES, can be found in Figure 3.

## 4. Discussion

In this review study, despite lacking a head-to-head comparison, we found the diagnostic utility of prenatal ES to be 69.0%, surpassing that of CMA manyfold. The reported diagnostic yield of CMA in detecting pathogenic CNVs in cases of suspected skeletal dysplasias was only 1.7–7.9% [12,23]. This would be due to the predominantly monogenic nature of skeletal dysplasias, which is undetectable with CMA. Our data suggested that there was a diversity of genetic defects identified in this cohort, and ES should be initiated as the routine prenatal management of cases suspected of skeletal dysplasias.

While 72% of the causative variants were de novo and are associated with conditions with autosomal dominant inheritance, 8.7% were inherited from either parent, which indicated that trio ES would favor data interpretation and genetic counseling. Among the causative variants identified in the analyzed cohorts, 70% occurred in one of the four recurring genes (*FGFR3*, *COL1A2*, *COL1A1*, and *COL2A1*). A retrospective study reported the diagnostic yield of the gene panel in suspected cases of skeletal dysplasias to be 42%. They utilized three different panels, encompassing 113 genes, 251 genes, and 374 genes, respectively [24]. The commonest genes identified were *COL2A1*, *FGFR3*, *COL1A1,* and *COL1A2*. In localities with limited resources, it may therefore be worthwhile to perform gene panel testing when typical sonographic features are found in the prenatal period. However, given the diversity of genes involved in skeletal dysplasias, there is a chance that the gene panel could return a negative, and extra time and resources are needed to go through whole-exome sequencing. Therefore, where resources allow, whole-exome sequencing might be more indicated and preferred over a targeted gene panel, especially when there are atypical sonographic features identified. Another circumstance is when there is a legal limit to the gestation of termination of pregnancy, where whole-exome sequencing would be more likely to identify any variants to further guide the counseling, especially among those fetuses presenting with isolated and nonisolated short long bones.

### 4.1. Diagnostic Yield

The overall additional diagnostic yield of prenatal ES in fetuses suspected of skeletal dysplasias is 57–88% across these studies, with an overall rate of 69.0%. This is in contrast with the two largest prospective studies that showed that the yield for skeletal abnormalities was just 15% and 24%, respectively. One of the reasons could be that in these studies, there was no mention of the actual case selection criteria. In both studies, women were eligible for inclusion whenever they were undergoing invasive testing for identified structural anomalies. If WES was performed in cases where the femur was just barely under the third centile without other abnormal features, the diagnostic rate would probably be remarkably lower. Nonetheless, in these two landmark papers, the additional diagnostic yield with WES was highest in the skeletal abnormalities group when compared to the other systems.

Even among the ten studies included, there was a notable variability in the inclusion criteria. Some included only short long bones as the only criteria, whereas some required there to be additional features such as abnormal curvature or that the femur length-to-abdominal circumference ratio be <0.16. In reality, a short femur is very often encountered by obstetricians, and only in a few situations would the cases be referred for invasive prenatal testing. Even though the cost of the next-generation technique has been reduced over the years, it is still not down to a level that can be easily self-financed by patients. Studies are usually aiming to discover novel genes that explain the phenomenon rather than to help the parents decide whether to continue with the pregnancy. Therefore, in selecting cases to go for ES, it is unsurprising to see a higher diagnostic rate when such highly selected cases were chosen to undergo ES, in contrast to those where parents could opt in for ES whenever invasive testing was performed, as in the two larger prospective studies. In these studies, there was no mention of the degree of bone shortening that defined the indication for ES.

A recent large systemic review and meta-analysis has shown the overall additional diagnostic yield across all fetal abnormalities to be 31% [25]. A subgroup analysis based on different systems showed the highest one to be skeletal dysplasias, at 53%. The authors also explained that case selection had a significant impact on the diagnostic yield.

At the time of drafting this study, another cohort study had been published from China, showing an additional diagnostic rate of 40.4% (38 out of 94 fetuses) with exome sequencing [26]. Similar to our study, it had shown that the diagnostic yield in fetuses with isolated short long bones was lower than in the fetuses with nonisolated short long bones but did not reach statistical significance. It also concluded that those with femur length below −4 SDs were more likely to have arrived at a particular genetic variant on exome sequencing than those with a femur length between −2 SD and −4 SD and are more likely to have a worse outcome. The authors explained that the relatively lower diagnostic yield compared to other studies is due to the selection bias and smaller sample size in other studies. However, that study only mentioned the selection criteria as “fetuses with short long bones with normal routine genetic tests”. As previously discussed, short long bones are a fairly common finding, and it depends on which cases are being referred for invasive testing or have gone through exome sequencing. It is pivotal for fetal-maternal medicine subspecialists to correctly identify cases where exome sequencing could potentially yield positive results to guide counseling.

In this study, we have demonstrated that in addition to short long bones, certain signs/features may alert obstetricians that there are likely to be additional findings with ES; hence, it should be considered alongside conventional karyotype and CMA. The two categories with the highest additional diagnostic yield are abnormal skull shape or macrocephaly and small chest. Other features that also have a high diagnostic yield include curved long bones, fractures or angulated long bones, reduced or abnormal ossification of bones, and multisystem anomalies. These features should therefore be examined in detail during a prenatal ultrasound when short long bones are identified, for the finding of such features could guide the counseling on whether exome sequencing should be performed. However, fetuses with absent limbs or phalanges, and those with abnormal joint positions, have a relatively lower additional diagnostic yield when performing ES. Although these are not described in classical cases of skeletal dysplasias, they can be found in certain types of skeletal dysplasias and are well described in the fetal medicine foundation.

### 4.2. Optimization of Prenatal ES in the Era of CMA

One problem yet to answer is determining the specific situations when ES should be performed at the same time as karyotype and CMA. With the increasing popularity and higher resolution of CMA, it is often performed as the first-tier test nowadays. However, the tests typically take 7–10 days to conclude, and the morphology scan is usually performed at 20–22 weeks, which pushes the time for CMA results to be available to roughly 21–23 weeks of gestation. In places where there is a legal limit to the termination of pregnancy for fetal anomalies (in our territory, it is 24 weeks of gestation), this gives no time for the discussion of prenatal ES. Obstetricians face the challenge of counseling such couples where there have been no genetic causes identified, and so are forced to explain the wide range of prognoses from constitutionally small to the lethal types of skeletal dysplasias. Some markers are used to aid in determining the lethality of the fetal anomaly(such as FL/AC <0.16 and small thoracic circumference); however, if the situation does not fit these criteria, the decision eventually lies in the parents’ hands, and they very often will choose to terminate the pregnancy given the uncertainties. The most difficult cases to counsel are not those with obvious lethal features of skeletal dysplasias but those with mild to moderate features and a normal CMA. As more and more novel genes have been discovered to explain these fetal anomalies, the application of ES should start to move from finding further novel genes to actually aiding prenatal counseling, as a molecular diagnosis made during the prenatal period could make a great impact on further management.

The International Society of Prenatal Diagnosis (ISPD) currently suggests that prenatal sequencing would be beneficial in situations where no genetic diagnosis was found after CMA and a clinical genetic expert review considers the phenotype suggestive of a possible genetic etiology, or when the “pattern” of multiple anomalies strongly suggests a single gene disorder. In the latter case, CMA should be run before or in parallel with prenatal exome sequencing. Based on this review, with the predominantly monogenic nature of skeletal dysplasias, exome sequencing should be performed after a negative CMA. In situations where there is a time constraint to the diagnosis (such as when there is a legal limit to the time of termination of pregnancy), WES should be considered to be the first-line investigation in conjunction with CMA.

### 4.3. Strengths and Limitations

As skeletal dysplasias are still considered a rare entity, most studies performed previously were of a small sample size. This review combines these studies systematically to arrive at an overall diagnostic yield. It applies the same stricter criterion to all studies and excludes all VUS. This is also the first study that investigates various additional features to determine specific sonographic features with the highest potential diagnostic yield.

The main limitation is the retrospective nature of most of these studies, and there could be selection bias in choosing the most severe cases to undergo ES. Also, 9 out of 10 studies were performed in China, so the applicability of the results could be affected based on the population demographics. The pregnancy outcomes were also not included in a large proportion of the studies.

## 5. Conclusions

Prenatal exome sequencing should be considered for cases with suspected fetal dysplasias with or without a negative karyotype or CMA. Certain sonographic features, such as an abnormal skull and small chest, indicate a potentially higher ES diagnostic yield.

## Figures and Tables

**Figure 1 genes-14-01203-f001:**
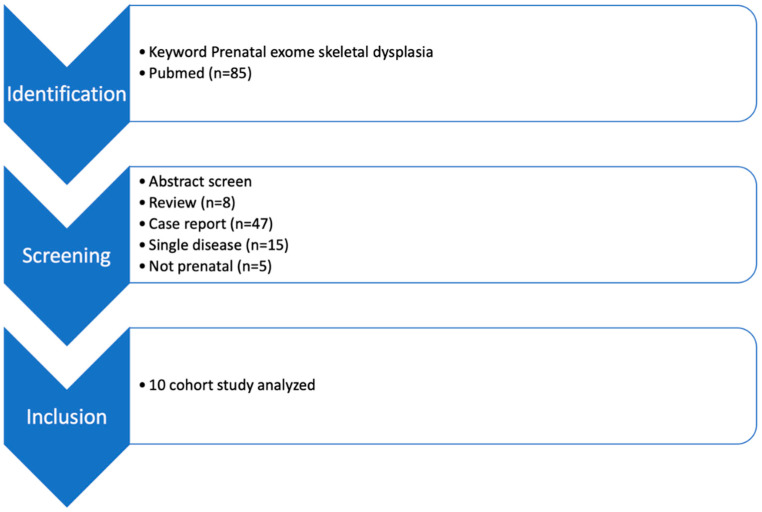
The workflow of this study.

**Figure 2 genes-14-01203-f002:**
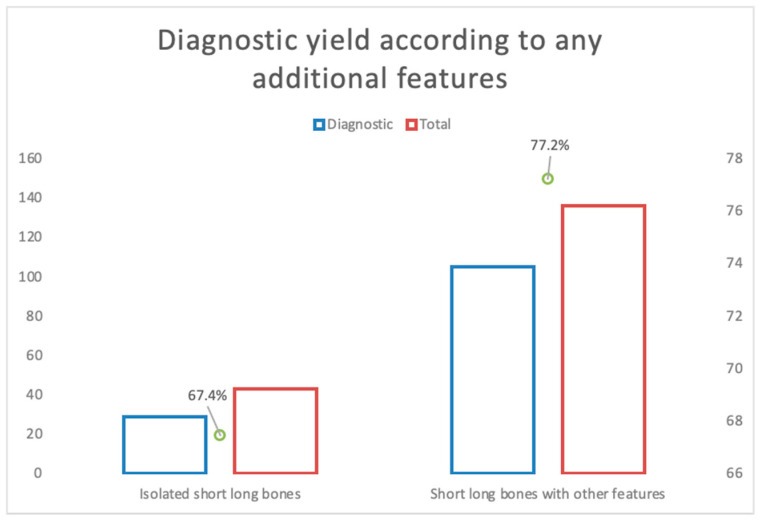
Comparison of additional diagnostic yield based on isolated or nonisolated short long bones.

**Figure 3 genes-14-01203-f003:**
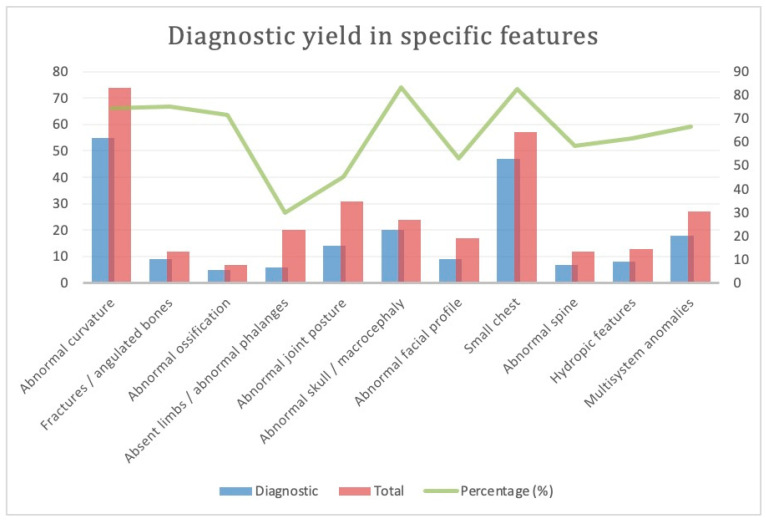
Diagnostic yield in prenatal cases with specific features. The graph describes the cases with individual sonographic features of skeletal dysplasias. The red bar describes the total cases of each individual category (see left Y axis). The blue bar represents the cases with causative variants found (see left Y axis), and the green line is the respective additional diagnostic yield (see right Y axis).

**Table 1 genes-14-01203-t001:** Summary of studies included in this study.

Authors	Year	Total Sample	Diagnostic Yield	Reference
Chandler et al.	2018	16	13 [81.3%]	[13]
Yang K et al.	2019	8	6 [75%]	[14]
Liu et al.	2019	28	16 [57.1]	[15]
Tang et al.	2020	8	6 [75%]	[16]
Han et al.	2020	26	23 [88.5%]	[17]
Peng et al.	2021	38	24 [63.2]	[18]
Tang H et al.	2021	15	10 [66.7]	[19]
Zhang X et al.	2021	27	19 [70.4]	[20]
Zhang L et al.	2021	55	35 [63.6]	[21]
Yang et al.	2022	5	4 [80%]	[22]
Total		226	156 [69.0%]	

## Data Availability

Available upon request.

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
