# Peer review of "Diagnostic Yield of Exome Sequencing in Fetuses with Sonographic Features of Skeletal Dysplasias but Normal Karyotype or Chromosomal Microarray Analysis: A Systematic Review"

_genes, 2023, doi:10.3390/genes14061203_

Round 1
Reviewer 1 Report
The authors present a meta-analysis of 226 fetuses evaluated in multiple studies that were reported in 85 manuscripts between 2013 to 2021 in PubMed. These fetuses were suspected to have a skeletal dysplasia based on imaging studies and received chromosome and chromosomal microarray studies reported as normal. Exome studies were then performed leading to a diagnosis in 69% of the patients. Most cases were the result of de novo gene variants, while 8.7% had inherited variants. The yield increased in cases where bone shortening was associated with other abnormalities. They recommend that exome studies should be considered in prenatal cases associated with bone dysplasias. The dysplasias were suspected in the presence of significant shortening of the long bones, abnormal shape of long bones, decreased echogenicity of the skeleton and absent extremities. Most cases were accounted by variants in collagen type I and type II as well as the FGR3 genes, and a small number were alkaline phosphatase and ciliopathy genes.
This a nice simple review as they were able to summarize the value of different genetic methodologies in the evaluation of a fetus with a suspected bone dysplasia. Obviously, the different studies, most of them from China, have approached the study of their cohorts by different criteria limiting the overall numbers. However, the value of the exome use is highlighted as an important tool in the evaluation of fetal anomalies including skeletal changes.
Minor comments:
Most of the comments are grammar related.
The authors refer to skeletal dysplasia. Given that this is a group that represents around 500 conditions I would rephrase to the plural, skeletal dysplasias. i.e. skeletal dysplasias are a group of rare genetic disorders,
For the sentence “The focus of this review is prenatal diagnosis of skeletal dysplasia, therefore studies with mixed pre and postnatal presentations were only included if it was possible to extract 89 the prenatal cases.” I am not sure that is clear of what is extracting prenatal cases, please rephrase.
Discussion: detecting pathogenic CNVs in setting of prenatal finding of 256 suspected skeletal dysplasia is reported to be 1.7 – 7.9%. Please rephrase, is not clear. The says “such research fundings” rather than findings. I have encountered numerous idiom problems in the discussion section and I would recommend that it should be revised by a native English speaker.
See comments in the main text
Reviewer 2 Report
Page 1
ABSTRACT
Lines 16-18
“However, routine genetic testing gives inadequate information regarding the disease”
The statement is not necessarily true. I would replace with “Next Generation Sequencing has rapidly revolutionized molecular diagnostic approaches in fetuses with structural anomalies”
Lines 25-26
“The majority of which were de novo variants, and 8.7% of the variants were inherited”
Reword as: “The majority of the molecular diagnoses involved de novo variants (72%), While 8.7% of cases were due to inherided variant.
Lines 26-27
“That of isolated short long bones and non-isolated cases were 67.4% and 77.2%”
The meaning of the sentence is not clear. Can it be reworded as “The incremental diagnostic yield of Exome Sequencing over CMA was 67.4% for isolated short long bones and 77.2% for non-isolated cases” ?
INTRODUCTION
Line 41
“the most common featuers of skeletal dysplasia detected…”
Replace “features of” with “forms of”
Page2
Line 43
“…and achondrogenesis”
Add a reference at the end of the sentence.
Line 62
“ISPD”
Please, write the full name before introducing the acronym.
Materials and Methods
As a general note, systematic reviews should have a formal registration before being undertaken or published. There are services that can provide retrospective registration too, like https://inplasy.com/
Page 3
Figure1
The figure should be presented in the Results section
While not strictly necessary, it would be better to use the template provided at http://prisma-statement.org/prismastatement/flowdiagram.aspx?AspxAutoDetectCookieSupport=1
More specifically, the “Screening” section should be more detailed
Lines 106-106
The full Pubmed string search with Boolean operators should be provided
Page4
Results
Line 164
The exclusion criteria and the number of papers excluded for each specific reason (as stated in Figure 1) should also be presented in the text.
Page 5
Lines 180-182
“Of which… parental origin”
Reword as “ Of the molecular diagnoses, 72% were due to de novo variants, and 8.7% were due to variants inherited from parental origin”
In addition, were all the pathogenic variants heterozygous/monoallelic? This should be specified in the text.
Line 194
“Of which”
Reword as “of these”
Page 7
Discussion
Lines 264-265
“Of Which… 70% of cases”
Reword as “Among the causative variants identified in the analysed cohorts, 70% occurred in one of 4 recurring genes (FGFR3, COL1A2, COL1A1, COL2A1).
Figure 2
This should be presented in the “results” section
Page 8
Line 295
“the inclusion criteria could vary a lot”
Reword as “there was notable variability in the inclusion criteria”
Lines 299-304
“Even though the cost…with the pregnancy”
While the statements are more than sensible, the sentences could use a thorough revision of English grammar and syntax.
Also, the economical considerations only apply to out-of-pocket health systems, and the application of exome sequencing in ongoing pregnancies for research purposes is not as common as a practice as it can be assumed from the paragraph.
Figure 3
The figure should be presented in the Results section. In addition, there should be a caption explaining how to read it.
Page 9
Line 318
“foetuses”
The American English spelling “fetuses” should be preferred
Line 334
“Role of prenatal ES in the era of CMA”
I would reword the title of the paragraph “Optimization
English language requires moderate revision
Round 2
Reviewer 2 Report
INTRODUCTION
Line 70
“Recently International Society for Prenatal Diagnosis”
Reword as : “Recently, the International Society for Prenatal diagnosis
MATERIALS AND METHODS
Lines 115-116
“We Search ((prenatal) AND (exome)) AND (skeletal) or ((prenatal) AND (exome)) AND (structural) in the Pubmed”
Reword as :” We searched the Pubmed database for ((prenatal) AND (exome)) AND (skeletal) or ((prenatal) AND (exome)) AND (structural)
Line 183
“only interrogated single disease”
Reword as: “only focused on a single condition, rather than skeletal dysplasias as a whole”
Lines 183-184
“5 studies were not prenatal samples”
Reword as : “5 studies did not include prenatal samples”
RESULTS
Line 185
“Of which,”
Reword as “Among these,”
Lines 202-204
“Of the 156 cases received molecular diagnoses, 134 (85.9%) were monoallelic / heterozygous variants, while 22 (14.1%) were homozygous / compound heterozygous variants.”
Reword as: “Of the 156 cases with molecular diagnoses, 134 (85.9%) were due tomonoallelic / heterozygous variants, while 22 (14.1%) were due tohomozygous / compound heterozygous variants”
Line 206
“or did not mention in the study”
Reword as “or the results of parental analysis were not mentioned in the studies”
Line 208
“They accounting for”
Reword as either “accounting for”, or as “which accounted for”
Line 214
“contributing”
The verb should be “contributing to”
Line 215
“The most common gene pathogenic variant found was in FGFR3”
Reword as “The gene harboring most of the Pathogenic variants in most of the cases was FGFR3”
Lines 217-218
“Of all the cases ,there were 14 cases (32.6%) ended up with TOP”
Reword as “ Of all the cases, 14 (32.6%) resulted in TOP”
Line 223
“There were 93 cases (68.4%) underwent TOP.”
Reword as “There were 93 cases (68.4%) undergoing TOP”
Line 226
“Of which”
Reword as “Among these, “
Line 283
Figure 3 caption
The caption should guide in more detail on how to read the symbols in the figure
DISCUSSION
Line 296
“and detected on the autosomal dominant genes”
Dominance is a relative attribute of phenotypes and alleles, not an absolute descriptor of genes. The sentence should be reworded as “associated with conditions with autosomal dominant inheritance”
Line 310
“and preferred than a targeted gene panel or CMA,”
Reword as “ and preferred to a targeted gene panel” or as “ preferred over a targeted gene panel” or as “preferred, rather than a targeted gene panel”. I would remove the statement on CMA in this sentence, as the comparison between ES and CMA is well discussed elsewhere in the paper and is not completeley appropriate in the present sentence.
Line 344
“of how short the long bones are that triggered the initiation of ES”
Reword as : “Of the degree of bone shortening that defined the indication for ES”
Line 411
“Strength and limitation”
The plural “strengths and limitations” should be preferred.
English language requires moderate editing (minor revision) and should be improved
